# Hospital Antibiotic Consumption before and during the COVID-19 Pandemic in Hungary

**DOI:** 10.3390/antibiotics13010102

**Published:** 2024-01-20

**Authors:** Roxána Ruzsa, Ria Benkő, Helga Hambalek, Erika Papfalvi, Dezső Csupor, Róbert Nacsa, Márta Csatordai, Gyöngyvér Soós, Edit Hajdú, Mária Matuz

**Affiliations:** 1Institute of Clinical Pharmacy, Faculty of Pharmacy, University of Szeged, 6725 Szeged, Hungary; benko.ria@med.u-szeged.hu (R.B.); hambalek.helga@szte.hu (H.H.); papfalvi.erika@med.u-szeged.hu (E.P.); csupor.dezso@szte.hu (D.C.); csatordai.marta@szte.hu (M.C.); soosgyongyver@szte.hu (G.S.); 2University Pharmacy Albert Szent-Györgyi Health Center, University of Szeged, 6725 Szeged, Hungary; 3Department of Emergency Medicine, Albert Szent-Györgyi Health Center, University of Szeged, 6725 Szeged, Hungary; 4Institute of Translational Medicine, Medical School, University of Pécs, 7624 Pécs, Hungary; 5Department of Internal Medicine Infectiology Unit, Albert Szent-Györgyi Health Centre, University of Szeged, 6725 Szeged, Hungary; horvathne.hajdu.edit@med.u-szeged.hu

**Keywords:** COVID-19, antibiotics, consumption

## Abstract

The aim of this study was to assess antibiotic use in the Hungarian hospital care sector during and before the pandemic. Aggregated systemic antibiotic (ATC: J01) utilisation data were obtained for the 2010–2021 period. Classifications and calculations were performed according to the WHO ATC/DDD index and expressed as DDD per 1000 inhabitants and per day (DID), DDD per 100 patient-days (DHPD) and DDD/discharge. A linear regression (trend analysis) was performed for the pre-COVID years (2010–2019) and a prediction interval was set up to assess whether the pandemic years’ observed utilisation fit in. Antibiotic utilisation was constant in DID before and during the pandemic (2019: 1.16; 2020: 1.21), while we observed a substantial increase in antibiotic use when expressed in DDD per 100 patient-days (2019: 23.3, 2020: 32.2) or DDD/discharge (2019: 1.83, 2020: 2.45). The observed utilisation level of penicillin combinations; first-, third- and fourth-generation cephalosporins; carbapenems; glycopeptides; nitroimidazoles and macrolides exceeded the predicted utilisation values in both pandemic years. Before the pandemic, co-amoxiclav headed the top list of antibiotic use, while during the pandemic, ceftriaxone became the most widely used antibiotic. Azithromycin moved up substantially on the top list of antibiotic use, with a 397% increase (2019: 0.45; 2020: 2.24 DHPD) in use. In summary, the pandemic had a major impact on the scale and pattern of hospital antibiotic use in Hungary.

## 1. Introduction

The coronavirus disease (COVID-19)-related pandemic exerted enormous pressure on health care systems around the world [1,2]. During the pandemic, elective procedures were delayed, and medical staff were reallocated to align hospital capacity with the demands of COVID-19 treatment. Studies from various countries also revealed that hospital admissions for non-COVID-19 patients, including medical and surgical cases, decreased significantly during the COVID-19 pandemic [3,4]. Restrictions and people’s fear of the virus could have also contributed to the decrease in hospital admissions. Studies on hospital antibiotic use during the pandemic years are scarce [5,6,7,8,9,10,11,12,13,14]. Most of the studies were from western European countries like Sweden, Switzerland, and Italy. From Central Europe, only one Croatian study has been published so far [8]. Most of these studies focused on the first year [7,9,10,12,13] or months [8,11,13] of the COVID-19 pandemic. Some studies focused on a single clinical centre [12,14], one involved a subset of hospitals in the data analysis [13], while others reported national-level data from hospitals [7,8,9,10,11]. These drug utilisation studies revealed an increase in hospital antibiotic use during the pandemic [7,8,9,10,12,13,14]. There was concern that the excessive prescription and improper use of antibiotics during the pandemic could worsen the already significant problem of antibiotic resistance. The impact of the COVID-19 pandemic on hospital antibiotic utilisation has not yet been investigated in Hungary. With this study, our objective was to evaluate the use of hospital antibiotics at the national level, during and before the COVID-19 pandemic in Hungary. We focused our analysis on both pandemic years (2020, 2021). 

## 2. Results

Table 1 shows the aggregated data on systemic antibiotic (ATC: J01) use in Hungarian hospitals.

When expressed in DDD per 1000 inhabitants and per day (DID), antibiotic utilisation was basically constant (2019: 1.16 vs. 2020: 1.21 DID; 4.3% increase). When expressed in DDD per 100 patient-days (DHPD) or DDD/discharge, considerable increases of 38% and 34% were observed, respectively (2019: 23.33 DHPD vs. 2020: 32.19 DPHD; 2019: 1.83 DDD/discharge vs. 2020: 2.45 DDD/discharge). 

In the first years of the pre-COVID period, the majority of systemic antibiotics used in Hungarian hospitals were oral formulations (Table 2). Later, a gradual increase in parenteral antibiotic use (2010: 7.3 DHPD vs. 2019: 10.8 DHPD) and a further sudden and significant increase during the pandemic years (2019: 10.8 DHPD vs. 2020: 17.4 DHPD, 61% increase) were observed. Parenteral antibiotics constituted 33.3% of total systemic antibiotic use in Hungarian hospitals in 2010 and 57.6% in 2021. (Table 2, Appendix A) In addition, the observed parenteral antibiotic use in the first year of the pandemic was much higher than the predicted value.

The antibiotics that were used the most during the pandemic years were third-generation cephalosporins (ATC: J01DD). Their use gradually increased in the pre-COVID years (from 2.26 DHPD in 2010 to 3.70 DHPD in 2019), while their use exerted a sharp increase during the pandemic (to 6.30 DHPD in 2020) (Table 2, Appendix A). The use of penicillin combinations with beta-lactamase inhibitors (ATC: J01CR) had been substantial and was stagnating in the pre-COVID years (2010: 5.31 DHPD vs. 2019: 5.04 DHPD), while it increased by 12% in 2020 (2020: 5.04 DHPD). Carbapenem consumption had been steadily increasing during the pre-COVID years and increased by 70% in 2020 (2019: 1.02 DHPD vs. 1.74 DHPD in 2020). With the exception of extended-spectra penicillin (J01CA) use in the year 2020 and second-generation cephalosporin (J01DB) use in both pandemic years, the observed use of all other beta-lactam antibiotic subgroups exceeded the predicted values (Table 2, Appendix A). 

In the pre-COVID years, fluctuations in the use of macrolides (ATC: J01FA) were observed. However, in the year 2020, a sharp increase in macrolide use was detected (from 1.84 DHPD in 2019 to 4.30 DHPD in 2020, a 133% increase), which largely exceeded the prediction interval (Table 2, Appendix A). The use of macrolides was moderated to 3.70 DHPD in the second pandemic year (2021), but they were still among the most used antibacterial subgroups. 

The use of quinolones (ATC: J01M) showed a decreasing trend in the pre-COVID years, from 5.00 DHPD in 2010 to 2.91 DHPD in 2019. A significant increase in their use was detected in 2020 (2019: 2.91 DHPD 2020: 3.69 DHPD), followed by a slight reduction in 2021 (3.49 DHPD) (Table 2). The observed values for the quinolone use fell within the prediction interval in both pandemic years.

Table 3 presents the top ten antibacterials used in different study years. Out of the top ten, six antibacterial drugs, namely co-amoxiclav, ciprofloxacin, cefuroxime, ceftriaxone, clarithromycin, and levofloxacin, were always present on the list. In the pre-COVID study years, co-amoxiclav remained the most used antibacterial agent, while during the pandemic years, ceftriaxone became the top antibacterial agent, followed by co-amoxiclav. In the first study year (2010) three, and in other years two fluoroquinolone agents were on the top list (see Table 3). The third most used antibacterial agent was cefuroxime in all highlighted years. Metronidazole continually creeped up on the top list and was the fourth most used agent in 2021. Azithromycin was not among the top 10 agents in the pre-COVID years but was in the fourth and fifth position in 2020 and 2021. Notably, cefazolin disappeared, while imipenem-cilastatin appeared on the top 10 list.

## 3. Discussion

Our study focused on the national-level hospital utilisation of antibiotics over the past ten years in Hungary, comparing the years before and during the COVID-19 pandemic. To the best of our knowledge, only Croatian researchers have conducted a similar study on hospital antibiotic use from Central Europe; all other publications were derived from western/northern European countries (Italy, England, Sweden, Switzerland, Poland, Spain) [7,9,10,11,12,13,14]. The present study reveals that antibiotic utilisation increased dramatically during COVID-19 in the Hungarian hospital care sector when using denominators related to hospital performance statistics. When expressed in DDD per 100 patient-days (DHPD), we observed a 38% increase (2019: 23.33 vs. 2020: 32.19DHPD), while when expressed in DDD per one discharge, we observed a 33.8% increase (2019: 1.83 vs. 2020: 2.45 DDD/discharge) in antibiotic use. Population-based metrics, i.e., DDD per 1000 inhabitants and per day (DID), revealed constant (2019: 1.16 vs. 2020:1.21 vs. 2021: 1.12 DID) antibiotic use in the Hungarian hospital care sector in all study years.

Investigations in other countries indicated different trends in hospital antibiotic use when using various outcome measures [7,8,9,10,11]. An Italian study found that when expressed as DDD per 1000 inhabitants per day (DID), there was only a 0.8% increase in hospital antibiotic use, while when expressing use in DDDs per 100 patient-days (DHPD), antibiotic utilisation in 2020 increased by 19.3% compared to the 2019 value [7]. In an English study, significant differences were also observed when applying different outcome measures (DID, DDD/1000 admissions) [9]. The total antibiotic consumption measured in DDD per 1000 inhabitants and per day (DID) reached its lowest point (−12.1%) in the pandemic’s first wave, yet when measured in DDD per 1000 hospital admissions, it showed an overall increase of 12% during the pandemic compared to pre-COVID years (with a doubling rate in April 2020 compared to April 2019 (7228 vs. 4681 DDD/1000 hospital admissions) [9]. Swiss colleagues also identified the difference in trends in hospital antibiotic use, depending on whether the values were expressed in DDD per 1000 inhabitants and per day (DID) or DDD per 100 patient-days (DHPD) [11]. The differences in trends using various outcome measures lies in the use of different denominators. The DDD per 1000 inhabitants and per day (DID) measurement unit has an annually standardised denominator (population size), while other outcome measures include performance indicators. Antibiotic utilisation was expressed in DDD per 100 patient-days (DHPD) or DDD/discharge as recommended by the World Health Organisation (WHO) and the handbook of drug utilisation research [15,16]. These outcome measures incorporate hospital patient turnover data. During the pandemic years, hospital admissions for non-COVID-19 patients, including medical and surgical cases, were significantly reduced during the COVID-19 pandemic, similar to data from other countries [3,4]. Performance indicators exerted the following decrease in Hungary during the pandemic: the number of hospital discharges decreased from 2,264,892 in 2019 to 1,761,372 in 2020 due to the postponement of elective procedures and the reduction in hospital admissions for less urgent medical cases [17,18], and in parallel, the number of registered patient days decreased significantly from 17,755,195 in 2019 to 13,386,617 in 2020. In summary, the bed occupancy rate was 71.84% in Hungarian hospitals in 2019, and it decreased to 55.02% in 2020 [18]. 

Although international evidence-based guidelines [19,20] have rejected the use of antibiotics in patients with mild, moderate, or severe COVID-19 infection without a suspected bacterial co-infection, the increased antibiotic use indirectly suggests that antibiotic utilisation was substantial in COVID-19 cases. Some studies have also shown that antibiotic use in hospitals was significantly higher, despite insufficient evidence for bacterial co-infection in a high percentage of hospitalised patients with COVID-19 [21,22].

The use of parenteral antibiotics also increased in Hungary; in the pre-COVID years, we observed a steady rise (2010: 7.3 DHPD vs. 2019: 10.8 DHPD), while during the pandemic, a sudden increase was observed (2020: 17.4 DHPD, 61% increase). The high and increased proportion of parenteral antibiotic use could also indirectly show the limited de-escalation due to the lack of a parenteral to per os switch.

Unlike the increasing trends for systemic antibiotic use observed in Hungary, Swiss colleagues found that the hospital antibiotic use remained relatively stable in 2020 compared to 2019 (+1.7%) when measured in DDD per 100 patient-days, while when expressed in DDD per 1000 inhabitants and per day (DID), they observed a decrease in use [11]. Narrowing down the focus on only the use of broad-spectrum antibiotics (meropenem, ertapenem, imipenem/cilastatin, aztreonam, cefepime, ceftazidim, piperacillin with tazobactam, ticarcillin), the Swiss study reported increasing use both in DDD per 1000 inhabitants and per day (DID) (+10.2%), as well as in DDD per 100 bed-days (DHPD) (+12.3%) [11]. We also detected an increase in the use of several broad-spectrum antibiotic groups, including third- and fourth-generation cephalosporins and carbapenems during the pandemic years, and we found that the observed utilisation values were significantly higher than predicted ones. The detected increased carbapenem use is a particularly serious problem considering the potential for further development and spread of antibacterial resistance [23]. The increased carbapenem use can be partly explained by the fact that during the pandemic, the number of nosocomial infections with ICU admission increased, for which carbapenems are often the first-line drugs. Furthermore, the annual incidence of health care-associated infections caused by third-generation cephalosporin-resistant (ESBL-producer) Klebsiella spp. per 100,000 patient days increased from 7.52 (in 2019) to 8.71 (in 2020) and increased further to 10.73 in 2021. For the treatment of infections caused by third-generation cephalosporin-resistant Klebsiella spp., treatment options include carbapenems [24,25,26]. As a consequence of increased carbapenem use, the proportion of carbapenem-resistant Klebsiella pneumoniae isolates in invasive samples increased recently (it was <1 % until 2021 and increased to 5.3% in 2022) according to the ECDC annual epidemiological report [27,28].

The use of third-generation cephalosporins in Hungarian hospitals increased significantly by 22% during the pandemic: from 5.04 DDD per 100 bed-days in 2019 to 6.13 DHPD in 2021. This rise can be explained by the fact that ceftriaxone was commonly used as a first-line agent in the empirical treatment of community-acquired pneumonia due to its convenient dosing regimen and low price, and this practice has been expanded to COVID-related pneumonia as well (when bacterial co- or superinfection was suspected). There was also a rise in the use of third-generation cephalosporins, particularly ceftriaxone, in a Spanish study, although that study included only one hospital and the *p* value did not reach statistical significance [14]. Furthermore, ceftriaxone use claimed the first place on the Hungarian top 10 list of antibiotic use during the pandemic years, surpassing the use of co-amoxiclav, which was the top antibiotic for many years. We also observed a significant increase in the hospital use of macrolides (ATC code: J01FA) during the pandemic, which more than doubled from 2019 to 2020 (2019: 1.84 vs. 2020: 4.30 DHPD), surpassing the predictive values considerably. Furthermore, azithromycin not only appeared in the top 10 list of antibiotics but also claimed the fourth position in 2020. In the early period of the COVID-19 pandemic, azithromycin use was recommended by international and national clinical guidelines based on its immunomodulatory and antiviral effects [20,29]. The first concern regarding the non-evidence-based use of azithromycin in hospitalised patients with COVID-19 was reported in the summer of 2020 [30], and it was re-confirmed in March 2021 that azithromycin had no beneficial effect in this patient group [31]. Similarly to other antibiotics, azithromycin use was only recommended for bacterial (co)-infections [30,31]. The WHO also did not recommend the use of azithromycin for COVID-19 treatment due to concerns about cardiotoxicity and the development of antibacterial resistance [19]. Many drug utilisation studies reported the increased use of macrolides or azithromycin during the COVID-19 pandemic in the hospital care sector [7,8,9,10,11]. Although the magnitude of increase was not as substantial as in Hungary, Croatian colleagues reported pronounced azithromycin use (+31%, from 4.8 to 6.3 DHPD) [8]. An Italian study also confirmed a substantial rise (+159.9%) in azithromycin use in the first semester of 2020 [7]. In a Hungarian study that analysed the impact of COVID-19 on ambulatory care antibiotic use, a significant increase in the use of azithromycin was also observed during the pandemic period. Hence, our data prove that suboptimal azithromycin use was not limited to the ambulatory care sector in Hungary [32]. 

A significant increase was also observed in the use of metronidazole from 0.86 DHPD in 2019 to 1.90 in 2020, which further rose to 2.05 DHPD in 2021. Since only aggregated drug utilisation data were available without linkage to indication, we can only presume that the increased use of metronidazole could result from the increased incidence of Clostridioides difficile infections. The Hungarian COVID-19 pandemic had a significant impact on the hospital epidemiology of infections caused by *C. difficile*. In 2020, a minor increase of 2.5% (2019: 5657 cases; 2020: 5800 cases), while in 2021, a substantial (8428 cases) increase of 45.3% was observed in the number of newly developed and reported health care-associated infections caused by *C. difficile*, which led to a significant increase in incidence and incidence density [24,25,26].

### Strengths and limitations

One of the main strengths of this study is the inclusion of both pandemic years (2020, 2021) in the analysis. Secondly, our dataset provides excellent (nearly 100%) coverage both in terms of Hungarian hospitals and systemic antibiotics. These data are also used for the yearly data submission of the European Center for Disease Prevention and Control (ECDC) and used in their annual reports; hence, validation is made regularly [33]. Some limitations might also apply to this research. We did not have access to antibiotic use data at the monthly or weekly level, which precluded the application of interrupted time serial analysis and the more detailed analysis of trends in antibiotic use. Secondly, we used an aggregated dataset which precluded data on indications, prescribers, or the individual exposure of patients. Therefore, we could not link antibiotic use to indication and could not evaluate the appropriateness of antibiotic use at the patient leve

## 4. Materials and Methods

This study examined antibiotic use between 2010 and 2021. Sales data (from wholesalers to hospitals) were obtained, and two main periods were determined: the pandemic period (2020–2021) versus antibiotic utilisation data from the previous 10 years (2010–2019). Data were compared in terms of quantity, composition, and trend. The dataset that we used provides complete coverage of Hungarian hospitals, including all public, church, and private hospitals.

Patient turnover statistics (number of patient-days and number of discharges) were obtained from the annual reports of the National Health Insurance Fund of Hungary (Hungarian acronym: NEAK) [18]. Population data were derived from the Hungarian Central Statistical Office’s reports [34]. 

Systemic antibacterials were classified according to the Anatomical Therapeutic Chemical (ATC) classification system (ATC: J01) of the World Health Organisation (WHO). Calculations were performed according to the WHO-defined Defined Daily Dose (DDD) methodology. DDD stands for the average daily maintenance dose for adults for the main indication of the active ingredient and corresponds to moderate-severity infection. Utilisation data were calculated using the WHO ATC/DDD index (version 2022) [35], and the results were reported as DDD per 100 patient-days (DHPD), DDD/1000 inhabitants per day (DID), and also in DDD/discharge as well. The day of admission and the day of discharge were counted together as one patient day. A linear regression (trend analysis) was performed for the pre-COVID years (2010–2019). The trend was determined by the regression coefficient (the average annual change) and the significance of the regression coefficient (*p*-value). A *p*-value of ≤0.05 was considered statistically significant. A prediction interval (with 95 % CI) was set up, and antibiotic use during the pandemic years (2020, 2021) was assessed to determine whether it fit in. Calculations were performed using R statistical software version 4.2.1 (R Core Team, Vienna, Austria). 

## 5. Conclusions

A significant increase in the use of systemic antibiotics was observed in Hungary during the pandemic years when expressing use in outcome measures that incorporate patient turnover statistics. This upward trend was confirmed for almost all antibiotic subgroups, including broad-spectra antibacterials. The pattern of use also changed, and parenteral antibiotics became dominant. The widespread use of macrolides (i.e., azithromycin) was detected during the pandemic, despite its unproven effectiveness for the treatment of COVID-19. Our results also highlight the importance of using different outcome measures.

Further research is needed to evaluate whether systemic antibiotic use has returned to pre-COVID levels or not in recent years. 

## Figures and Tables

**Table 1 antibiotics-13-00102-t001:** Antibiotic use in pre-COVID and pandemic years expressed in different outcome measures.

	Years	DDDs *	DDD/1000 Inhabitants/Day (DID)	DDD/ 100 Patient-Days (DHPD)	DDD/ Discharge
Pre-COVID years	2010	4,363,348	1.19	21.97	1.77
	2011	3,838,936	1.05	19.45	1.54
	2012	3,947,868	1.09	20.61	1.66
	2013	3,866,325	1.07	20.14	1.61
	2014	4,019,441	1.11	20.96	1.66
	2015	4,016,330	1.12	21.53	1.68
	2016	3,796,523	1.06	20.33	1.59
	2017	4,005,607	1.12	22.01	1.75
	2018	3,997,055	1.12	22.16	1.76
	2019	4,142,182	1.16	23.33	1.83
Pandemic years	2020	4,308,489	1.21	32.19	2.45
	2021	3,989,643	1.12	33.65	2.42

* DDD: Defined Daily Doses.

**Table 2 antibiotics-13-00102-t002:** Trend analysis of antibiotic use expressed as DDD per 100 patient-days (DHPD).

		Pre-COVID Years’ Antibiotic Use Expressed in DDD per 100 Patient-Days (DHPD)										Pandemic Years’ Antibiotic Use Expressed in DDD per 100 Patient-Days (DHPD)		Change from 2019 to 2020 (%) *			Included in the Prediction Interval	
ATC Code	Name of Antibiotic Groups	2010	2011	2012	2013	2014	2015	2016	2017	2018	2019	2020	2021		Coefficient (Trend 2010–2019)	*p* Value (Trend 2010–2019)	2020	2021
J01		21.97	19.45	20.61	20.14	20.96	21.53	20.33	22.01	22.16	23.33	32.19	33.65	37.96	0.238	0.055	no **	no
J01	Parenteral antibiotics	7.3	7.5	8.3	8.3	9.0	9.2	9.4	10.5	10.7	10.8	17.4	19.4	61.31	0.411	0.000	no	no
J01A	Tetracyclines	0.60	0.54	0.58	0.60	0.62	0.68	0.54	0.59	0.64	2.05	1.49	1.90	−27.23	0.083	0.103	yes	yes
J01CA	Penicillins with extended spectrum	0.47	0.50	0.38	0.36	0.38	0.29	0.36	0.32	0.38	0.39	0.45	0.49	15.12	−0.012	0.083	yes	no
J01CE (CE)	Beta-lactamase-sensitive penicillins	0.13	0.08	0.08	0.08	0.03	0.01	0.01	0.02	0.01	0.01	0.06	0.06	402.94	−0.013	0.000	no	no
J01CR	Penicillin combinations including beta-lactamase inhibitors	5.31	4.94	5.25	4.97	5.25	5.05	4.83	5.11	5.03	5.04	5.69	6.13	12.90	−0.019	0.287	no	no
J01DB	First-generation cephalosporins	0.35	0.41	0.46	0.48	0.54	0.55	0.58	0.63	0.68	0.74	0.89	1.02	21.24	0.040	0.036	no	no
J01DC	Second-generation cephalosporins	2.15	1.76	1.83	1.95	1.95	2.06	2.03	2.19	2.20	2.23	2.34	2.14	5.04.	0.036	0.004	yes	yes
J01DD	Third-generation cephalosporins	2.26	2.38	2.65	2.70	2.75	2.97	2.82	3.19	3.40	3.70	6.30	6.99	70.46	0.142	0.000	no	no
J01DE	Fourth-generation cephalosporins	0.02	0.02	0.02	0.02	0.01	0.02	0.02	0.03	0.01	0.03	0.06	0.06	78.35	0.000	0.541	no	no
J01DH	Carbapenems	0.41	0.45	0.52	0.60	0.67	0.76	0.79	0.88	0.98	1.02	1.74	2.03	70.89	0.071	0.000	no	no
J01E	Sulphonamides and trimethoprim	0.70	0.58	0.61	0.63	0.61	0.67	0.67	0.68	0.69	0.68	0.78	0.86	14.59	0.007	0.140	yes	no
J01FA	Macrolides	2.17	1.30	1.44	1.32	1.43	1.68	1.34	1.64	1.70	1.84	4.30	3.70	133.68	0.007	0.836		
J01FF	Lincosamides	0.87	0.78	0.83	0.74	0.84	0.77	0.71	0.74	0.75	0.73	0.84	0.82	14.73	−0.013	0.020		
J01G	Aminoglycosides	0.67	0.63	0.59	0.57	0.58	0.57	0.55	0.55	0.34	0.39	0.67	0.61	69.58	−0.029	0.001	no	no
J01M	Quinolones	5.00	4.20	4.13	3.93	3.91	4.08	3.94	4.04	3.77	2.91	3.69	3.49	26.97	−0.134	0.006	yes	yes
J01XA	Glycopeptide antibacterials	0.23	0.30	0.33	0.30	0.34	0.37	0.27	0.44	0.46	0.39	0.63	0.74	62.04	0.018	0.010	no	no
J01XD01	Imidazole derivates (metronidazole)	0.51	0.57	0.66	0.60	0.71	0.72	0.75	0.81	0.86	0.88	1.90	2.05	115.42	0.040	0.000	no	no
Other ***		0.10	0.04	0.25	0.29	0.33	0.29	0.12	0.15	0.26	0.30	0.35	0.54	17.30				

* Calculation of the relative change in use from 2019 to 2020: [(Δ(DHPD_2020_ − DHPD_2019_)/DHPD_2019_] 100. ** No: exceeded the prediction interval. *** Other (ATC groups): J01DI02; J01DI54; J01XB01; J01XE01; J01XX01; J01XX04; J01XX08; J01X.

**Table 3 antibiotics-13-00102-t003:** Consumption of the top 10 systemic antibiotics in the years highlighted, expressed as DDD per 100 patient-days (DHPD).

2010						2019				
	Group of ATC	Systemic INN *	DDD per 100 Patient-Days	%	cum ** %	Group of ATC	Systemic INN *	DDD per 100 Patient-Days	%	cum ** %
1.	J01CR02	Co-amoxiclav	5.1	23.3	23.3	J01CR02	Co-amoxiclav	4.7	20.0	20.0
2.	J01MA02	Ciprofloxacin	2.8	12.9	36.2	J01DD04	Ceftriaxone	3.3	14.3	34.3
3.	J01DC02	Cefuroxime	2.1	9.5	45.7	J01DC02	Cefuroxime	2.2	9.5	43.7
4.	J01DD04	Ceftriaxone	1.6	7.2	52.9	J01AA02	Doxycycline	2.0	8.6	52.4
5.	J01FA09	Clarithromycin	1.3	6.1	59.0	J01MA02	Ciprofloxacin	1.4	6.1	58.4
6.	J01MA12	Levofloxacin	1.0	4.3	63.3	J01FA09	Clarithromycin	1.4	5.9	64.4
7.	J01FF01	Clindamycin	0.9	4.0	67.3	J01MA12	Levofloxacin	1.0	4.1	68.5
8.	J01FA10	Azithromycin	0.8	3.4	70.7	J01XD01	Metronidazole	0.9	3.8	72.2
9.	J01EE01	Sulfamethoxazol-trimethoprim	0.7	3.2	73.9	J01DB04	Cefazolin	0.7	3.2	75.4
10.	J01MA14	Moxifloxacin	0.6	2.6	76.5	J01FF01	Clindamycin	0.7	3.1	78.5
Pandemic years										
**2020**						**2021**				
	**Group of ATC**	**Systemic INN ***	**DDD per** **100 patient-days**	**%**	**cum ** %**	**Group of ATC**	**Systemic INN ***	**DDD per** **100 patient-days**	**%**	**cum ** %**
1.	J01DD04	Ceftriaxone	5.9	18.2	18.2	J01DD04	Ceftriaxone	6.5	19.3	19.3
2.	J01CR02	Co-amoxiclav	5.1	15.9	34.1	J01CR02	Co-amoxiclav	5.4	15.9	35.2
3.	J01DC02	Cefuroxime	2.3	7.2	41.3	J01DC02	Cefuroxime	2.1	6.2	41.4
4.	J01FA10	Azithromycin	2.2	7.0	48.3	J01XD01	Metronidazole	2.0	6.1	47.5
5.	J01FA09	Clarithromycin	2.0	6.4	54.6	J01FA10	Azithromycin	2.0	6.0	53.5
6.	J01XD01	Metronidazole	1.9	5.9	60.5	J01AA02	Doxycycline	1.8	5.2	58.7
7.	J01MA12	Levofloxacin	1.5	4.8	65.3	J01FA09	Clarithromycin	1.7	5.0	63.7
8.	J01MA02	Ciprofloxacin	1.5	4.6	69.9	J01MA12	Levofloxacin	1.6	4.8	68.5
9.	J01AA02	Doxycycline	1.4	4.4	74.4	J01MA02	Ciprofloxacin	1.3	3.8	72.3
10.	J01DH51	Imipenem + cilastatin	1.0	3.1	77.4	J01DH51	Imipenem + cilastatin	1.0	3.0	75.4

* INN: International Nonproprietary Names. ** cum: cumulative.

## Data Availability

Data are available from the corresponding author upon reasonable request.

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
