# Peer review of "Hospital Antibiotic Consumption before and during the COVID-19 Pandemic in Hungary"

_antibiotics, 2024, doi:10.3390/antibiotics13010102_

Round 1

Reviewer 1 Report

Comments and Suggestions for Authors

Here is a summary of the merits, pitfalls, and areas for improvement in this study on hospital antibiotic use during the COVID-19 pandemic in Hungary:

Merits:

- Compares antibiotic use before (2010-2019) and during (2020-2021) the COVID pandemic using national-level data in Hungary

- Uses multiple measures to quantify antibiotic use: DID, DHPD, DDD/discharge

- Performs trend analysis on pre-pandemic years to predict antibiotic use and assess if pandemic years exceeded predictions

- Provides a breakdown of use by antibiotic subgroup and top 10 antibiotics

Areas for Improvement:

Methods:

- More justification is needed for why the 2010-2021 timeframe was chosen [no major changes to provide more context?]

- Add information about data source(s) and what is included (public and private hospitals?)

Results:

- Table 1: Define abbreviations in the table header (DID, DHPD); add units in the table

- Table 2: Justify the choice of antibiotics highlighted; define abbreviations (SMT)

- Add monthly/quarterly breakdowns of use to see the impact over the pandemic timeline

Discussion:

- More clearly highlight the most important and novel findings in the first paragraph

- Relate findings on azithromycin overuse to guidelines at the time

- Discuss limitations: aggregated data precludes evaluation of appropriateness of use

Conclusions:

- Simplify the key message in the first sentence to focus on the impact on scale and patterns

- Note that findings may not apply to ambulatory antibiotic use

Other:

- Reduce abbreviations for readability (DHPD, DDD)

- Improve clarity of English grammar and wording throughout

- Table formatting could be improved (borders, text wrapping)

The study provides useful national-level data showing antibiotics were overused in Hungarian hospitals during the COVID-19 pandemic, raising concerns about resistance. Additional temporal and clinical details would further contextualize the findings and significance.

Comments on the Quality of English Language

Minor editing of the English language is required.

Author Response

Response to Reviewer 1.

We thank the Reviewer for the devoted time and for the critics which enabled to develop our manuscript.

The changes are highlighted in the manuscript

Comment 1: More justification is needed for why the 2010-2021 timeframe was chosen [no major changes to provide more context?]

Response: We considered this 10 year period sufficient to observe trends before the pandemic. Moreover we intended to cover both pandemic years for better understanding of AB use during COVID-19.

Comment 2: Add information about data source(s) and what is included (public and private hospitals?)

Response: This information is now provided clearly in the methods section.

Comment 3: Table 1: Define abbreviations in the table header (DID, DHPD); add units in the table

Response: Table headers were complemented with these informations.

Comment 4: Table 2: Justify the choice of antibiotics highlighted; define abbreviations (SMT)
Response: All ATC 2 level antibiotic subgroups are highlighted: For the two big beta lactam group: penicilins and cephalosporins we also showed the ATC 3 level (e.g cephalosporin generations).

Comment 5: Add monthly/quarterly breakdowns of use to see the impact over the pandemic timeline

Response: We agree that monthly/quaterly data would have given more insight, but our dataset is limited to annual records, providing us annual consumption/utilisation values.

Comment 6: More clearly highlight the most important and novel findings in the first paragraph

Response: We tried to rephrase the the first paragraph of the discussion to highlight more the main findings

Comment 7: Relate findings on azithromycin overuse to guidelines at the time

Response: No official Hungarian guideline is/was available on the website of Hungarian guidelines as there was no time to develope this. Therefore we refered to the WHO issued guidelines, which were translated into Hungarian documents and served as guide for COVID-19 treatment. Prescribers also used international databases such us Uptodate.

Comment 8: Discuss limitations: aggregated data precludes evaluation of appropriateness of use

Response: We agree with this remark and updated the limitation accordingly. However, some patterns (increasing use of macrolides) suggest suboptimal prescribing habits. Drug specific quality indicators are used for such purpose (to assess appropriateness) widely.

Comment 9: Simplify the key message in the first sentence to focus on the impact on scale and patterns

Response: We tried to rephrase the first sentence of the conclusion.

Comment 10: Note that findings may not apply to ambulatory antibiotic use

Response: We agree, our data only pertains to hospital care and cannot be extrapolated to the AC sector. We made the related sentence more clear to avoid confusion.

Comment 11: Reduce abbreviations for readability (DHPD, DDD)

Response: We tried to decrease the number of abbreviations in the manuscript text.

Comment 12: Improve clarity of English grammar and wording throughout

Response: We involved AI to improve the English of the manuscript.

Comment 13: Table formatting could be improved (borders, text wrapping)

Response: We reformated the tables.

Yours sincerely,
Roxána Ruzsa

Reviewer 2 Report

Comments and Suggestions for Authors

Please change 'The 2019 coronavirus' to Coronavirus diseases of 2019 (COVID-19), then you can use only abbreviations afterwards. 

You can add possible reasons of less number patients were visiting due to lockdown or any fear of getting COVID-19 besides HCWs reallocations etc.

An important point that need authors clarification that is they used antibiotics sale data and how they were assuming this as an antibiotics consumption/utilization? Although antibiotics sale data is a strong evidence that indicate antibiotics use/consumption during a particular time but you may indicate that evidences here in introduction by stating that you are relating antibiotics sale data to indicate its consumption etc. 

How rise in antibiotics use data will have impact on the health system? how it is contributing in the development of AMR in the country. What are current guidelines to inhibit excessive antibiotics use particularly in COVID-19 pandemic.

Methods section needs extensive revision like is the study involved antibiotics use in the whole country? I am not convinced by the data source indicating antibiotics sale nor authors were able to provide it in a way that reviewers/readers can get it. 

Data analysis also need revision like authors were unable to provide the population under consideration, how they differenciate dischage patients from admitted patients just looking at antibiotics sale data?

Comments on the Quality of English Language

The study required extensive English language editing. 

Author Response

Response to Reviewer 2.

We thank the devoted reviewer for the time and the critics that enabled us to develop our manuscript.

The changes are highlighted in the manuscript

Comment 1: Please change 'The 2019 coronavirus' to Coronavirus diseases of 2019 (COVID-19), then you can use only abbreviations afterwards. 

Response: Thank you for your remark. We introduced the abbreviation in the manuscript in the first sentence of the introduction.

Comment 2: You can add possible reasons of less number patients were visiting due to lockdown or any fear of getting COVID-19 besides HCWs reallocations etc.

 Response: Thank you for your remark. We agree that all these factors could contribute to the decreased number of hospital admissions. We complemented the sentence in the introduction.

Comment 3: An important point that need authors clarification that is they used antibiotics sale data and how they were assuming this as an antibiotics consumption/utilization? Although antibiotics sale data is a strong evidence that indicate antibiotics use/consumption during a particular time but you may indicate that evidences here in introduction by stating that you are relating antibiotics sale data to indicate its consumption etc. 

Response: Aggregate drug utilisation studies can utilise sales or reimbursement data to quantify drug use (drug consumption/drug utilisation) (ref: Elseviers et al. Drug Utilisation Research: Methods and Applications, Wiley, 2016). In Hungary, reimbursement data is only available for ambulatory care. Sales data provides a valid and slight overestimation of real drug use. Data obtained for this study was also utilised to provide the annual compulsary data submission to ECDC, which has proved validity.

Comment 4: How rise in antibiotics use data will have impact on the health system? how it is contributing in the development of AMR in the country. What are current guidelines to inhibit excessive antibiotics use particularly in COVID-19 pandemic.

Response: You first question might imply a future study. However, to provide some ’insight’ we added an example on this in the discussion section based on annual AMR reports published by ECDC (EARS-Net annual reports). ECDC AMR data has already proved increase in AMR rates for Hungary. Unfortunately, national level guidelines are mainly available for the ambulatory care sector in Hungary. During the COVID pandemic there were several guidelines. However these are not updated or issued any more.

Comment 5: Methods section needs extensive revision like is the study involved antibiotics use in the whole country? I am not convinced by the data source indicating antibiotics sale nor authors were able to provide it in a way that reviewers/readers can get it. 

Response: The study used national level aggregated data. Drug and population coverage was excellent so the whole country is covered by the data and all antibiotic sales to hospitals are included (the only exemption might be donated medicines which are marginal). We already included this information in the section on strengths. The validity of our dataset (including drug and hospital coverage) has been checked by ECDC, as sales data used in this study is also used in the annual data submission to ECDC.

Comment 6: Data analysis also need revision like authors were unable to provide the population under consideration, how they differenciate discharge patients from admitted patients just looking at antibiotics sale data?

Response: We cannot understand your question clearly. There was no need to differentiate patients. Discharge data was derived from public hospital turnover statistics. Discharge data was used as a dominator to quantify antibiotic utilisation in a standardized way recommended by WHO and also in the textbook on drug utilisation research (Elseviers et al: Drug Utilisation Research: Methods and Applications, Wiley, 2016, chapter 26).

Yours Sincerely,
Roxána Ruzsa

Reviewer 3 Report

Comments and Suggestions for Authors

This is a very interesting study on the use (increase of it) of antibiotics during the years of the pandemic reflecting a very serious problem with which the medical community will be confronted.

Two minor issues

1. The authors might need to comment on the increase of use of metronidazole and its potential causes

2. Data regarding national antimicrobial resistance changes would be very informative.

Comments on the Quality of English Language

Minor language editing

eg Line 121 expressingin

Author Response

Response to Rewiever 3.

We thank the devoted reviewer for the time and the critics that enabled us to develop our manuscript.

The changes are highlighted in the manuscript

Comment 1: The authors might need to comment on the increase of use of metronidazole and its potential causes.

Response: Our data is not linked to indication but we assume that increased metronidazole use was a consequence of C. diff infections. We added a paragraph on this in the discussion.

Comment 2: Data regarding national antimicrobial resistance changes would be very informative.

Response: Thank you very much for your comments. We added some data in the discussion section for better interpretation of related of AMU data.

Yours sincerely,
Roxána Ruzsa

Reviewer 4 Report

Comments and Suggestions for Authors

Ruzsa et al report on their evaluation of antibiotic consumption in Hungary before and during the COVID-19. The study has considerable public health significance, being one of the few to extensively document antibiotic consumption patterns prior to and during the COVID-19 pandemic. The background information provided by the authors is also sufficient. The novelty of the study has also been clearly articulated through highlighting the scope of previous studies on the subject. Moreover, the methods are adequately described, and the results clearly presented and discussed. The references cited are also appropriate, and the conclusions are in line with the results and meet the objectives of the study.

One major issue with the manuscript has to do with clarity of presentation. I recommend that the authors pay particular attention to this issue, and make use of a professional English language editing service.

Comments on the Quality of English Language

Moderate edits are needed.

Author Response

Response to Reviewer 4.

We thank the Reviewer for the helpful comments to which we reply as follows. Thank you also for encouraging us in our work!

Comment 1: One major issue with the manuscript has to do with clarity of presentation. I recommend that the authors pay particular attention to this issue, and make use of a professional English language editing service.

Response: Thank you for your feedback. We appreciate your suggestion regarding the clarity of presentation. The correctness of the language has been completely revised and improved.

Yours sincerely,
Roxána Ruzsa

Round 2

Reviewer 2 Report

Comments and Suggestions for Authors

Thank you for addressing all the concerns and congratulations on this work

Comments on the Quality of English Language

Thank you for the coordination and organizing the review process. I think authors have adequately addressed the comments therefore I endorse this study publication.